# Nickel Oxide Films Deposited by Sol-Gel Method: Effect of Annealing Temperature on Structural, Optical, and Electrical Properties

**DOI:** 10.3390/ma15051742

**Published:** 2022-02-25

**Authors:** Tatyana Ivanova, Antoaneta Harizanova, Maria Shipochka, Petko Vitanov

**Affiliations:** 1Central Laboratory of Solar Energy and New Energy Sources, Bulgarian Academy of Sciences, Tzarigradsko Chaussee 72, 1784 Sofia, Bulgaria; tonyhari@phys.bas.bg (A.H.); vitanovpetko@yahoo.co.uk (P.V.); 2Institute of General and Inorganic Chemistry, Bulgarian Academy of Sciences, Acad. G, Bonchev St., bl. 11, 1113 Sofia, Bulgaria; shipochka@svr.igic.bas.bg

**Keywords:** sol-gel technology, NiO, thin films, optical properties, electrical properties

## Abstract

In our study, transparent and conductive films of NiO_x_ were successfully deposited by sol-gel technology. NiO_x_ films were obtained by spin coating on glass and Si substrates. The vibrational, optical, and electrical properties were studied as a function of the annealing temperatures from 200 to 500 °C. X-ray Photoelectron (XPS) spectroscopy revealed that NiO was formed at the annealing temperature of 400 °C and showed the presence of Ni^+^ states. The optical transparency of the films reached 90% in the visible range for 200 °C treated samples, and it was reduced to 76–78% after high-temperature annealing at 500 °C. The optical band gap of NiOx films was decreased with thermal treatments and the values were in the range of 3.92–3.68 eV. NiO_x_ thin films have good p-type electrical conductivity with a specific resistivity of about 4.8 × 10^−3^ Ω·cm. This makes these layers suitable for use as wideband semiconductors and as a hole transport layer (HTL) in transparent solar cells.

## 1. Introduction

Nickel oxide (NiO) is a remarkable semitransparent semiconducting material with a cubic rock salt-like structure with octahedral Ni^2+^ and O^2^^−^ sites [1]. Nickel oxide is one of the interesting p-type wide band gap semiconductors with excellent properties, including high chemical and mechanical stabilities, extreme stability in a broad range of temperatures, and a direct band gap [2]. NiO is known to be a nontoxic material, especially compared to other transition semiconductors [3]. Due to these unique properties, NiO_x_ thin films become multipurpose materials in various applications such as electrochromic (EC) applications [4]. NiO manifests high optical contrast, fast switching times, good cycling, and anodic coloration, which is very promising for a counter electrode layer in complementary EC devices [5,6,7], supercapacitors [8], as a hole transport layer of perovskite solar cells [9], highly sensitive gas sensors [2], transparent conducting oxides, spin-valve giant magnetoresistive sensors, chemical sensor applications, Schottky diodes, and metal oxide field-effect transistors (MOSFETs) [10].

Usually, it is difficult to achieve stoichiometric NiO films and the stoichiometric NiO possesses a high electrical resistivity of 10^13^ Ω·cm [11]. Its electrical properties depend on the Ni and O lattice defects [2]. Nonstoichiometric nickel oxide (NiO_x_) has been attracting attention due to the characteristics of its p-type transparent conductive layer [12]. Recently, NiO films have attracted scientific research, as they are investigated for application in semitransparent solar cells and transparent photovoltaic solar cells (TPCs) [13,14,15]. Transparent and semitransparent photovoltaics initiate new fields of applications, such as energy harvesting on large windows or glass roofs, fields that are inaccessible with conventional technologies [16]. These solar cells are attractive due to their optical transparency and the greater flexibility of installation location that it permits. They also absorb ultraviolet (UV) light of the solar spectrum, which is harmful to humans [13]. Metal oxide-based solar cells have high potential to be used, as they have advantages due to excellent chemical stability and minor toxicity. These solar cells are based on the heterojunction p-n structures [17]. NiO has been investigated as a p-type layer in p/n junctions heterostructures for all-oxide optoelectronic applications, where the n-type layer for forming the heterostructure has been TiO_2_, ZnO, Ga_2_O_3_, etc. [18]. For example, NiO and ZnO are found to be available, stable, and nonexpensive, and can be fabricated by various methods on inexpensive substrates at low process temperatures.

Deposition of NiO thin films can be achieved by a variety of methods, including magnetron sputtering, pulsed laser deposition, electron-beam evaporation, chemical vapour deposition (CVD) and atomic layer deposition, thermal deposition, electrodeposition, spray pyrolysis, etc. [19,20]. Among these, sol-gel technology is widely used due to its relative simplicity, low cost, depositing uniform coatings, control of film thickness, and reproducibility [21,22].

It has been reported that NiO/ZnO heterojunction can lead to realization of all-transparent metal oxide photoelectric devices with transparent solar cell efficiency (6%), current density (2.7 mA/cm^2^) and open circuit voltage of 532 mV [23].

Sol-gel method for depositing thin layers is widely used, owing to its simplicity, and low cost.Unlike other technological methods, it is not a vacuum process and does not require expensive technological equipment. This paper presents studies on thin layers of nickel oxide obtained by the sol-gel method. With this approach it is possible to obtain non-stoichiometric nickel oxide. The stoichiometric form of NiO is an insulator with a very low intrinsic conductivity. At NiOx due to Ni vacancies, some Ni^2+^ ions have to be converted to Ni^3+^ ions in order to maintain the electrical neutrality in the structure, and the created Ni^3+^ ions take charge of conduction in NiO. Non-stoichiometric nickel oxide (NiO_x_) is attracting growing attention due to its characteristics of p-type transparent conductive layer. Good results have been obtained employing NiOx as hole transport layer (HTL) in organic and perovskite solar cells (PSCs). NiO_x_ films as HTL are reported to be able to replace commonly-used PEDOT:PSS or spiro-MeOTAD HTL for high efficiency performance PSCs [24]. 

In our study, we aimed to achieve a technological process for deposition thin films of p-type NiO_x_ which is compatible with integration into technology for wide-band heterojunction transparent solar cells. NiO_x_ films were deposited by sol-gel technology on Si and glass substrates. Uniform and smooth NiO_x_ films were spin-coated on glass and silicon substrates. The impact of the annealing temperatures on optical, structural, and electric properties of nickel oxides films was studied.

## 2. Materials and Methods

### 2.1. Thin Films Preparation

The sol solution synthesis and the thin film deposition is schemed in Figure 1. The sol solution for NiO film deposition was prepared from Ni acetate (Ni(CH_3_COO)_2_·4H_2_O) (Alfa Aesar, Karlsruhe, Germany, 98%+ pure) as precursor and dissolved in absolute ethanol (Merck, Darmsstadt, Germany, absolute for analysis) to reach 0.4 M concentration. The solution was stirred on a magnetic stirrer (type ARE, Velp Scientifica s.r.l., Usmate, Italy) at 60 °C for 30 min. Monoethanolamine (MEA, Fluka AG, Buch, Switzerland, 98%) was used as a complexing agent in order to prevent precipitation. The molar ratio MEA/Ni was fixed to 1. After adding MEA, the solution was treated again on a magnetic stirrer at 60 °C for 15 min. The solution was then ultrasonically (ultrasonic cleaner, UST 2.8-100, Siel Ltd., Gabrovo, Bulgaria) treated at 40 °C for 15 min. The sol solution had been aged for 24 h. The obtained Ni sol solution was found to be homogeneous and transparent without precipitations and with good film forming properties.

NiO_x_ thin films were spin coated on preliminary cleaned substrates (Spin coater P 6708, PI-KEM Limited, Staffordshire, UK). The substrates used for structural and surface chemistry studies were silicon wafers (FZ, p-type, resistivity 4.5–7.5 Ω, orientation ˂100˃), and for optical characterization, glass substrates were used. The substrates were preliminarily cleaned in acetone in an ultrasonic bath at 45 °C, ultrasonically treated in ethanol (60 °C), and finally washed in distilled water. After the coating solution was dispensed onto the substrates, the substrates were centripetally accelerated to 2000 rpm and spun at this rotational speed for 30 s in ambient air. After the spinning process, the films were dried at 100 °C for 10 min in order to remove organic residuals (preheating temperature procedure). The high temperature annealings were performed at the temperatures of 200, 300, 400, and 500 °C for 1 h in air with controlled constant heating and cooling rates of 10 °C/min in a chamber furnace (Tokmet—TK Ltd., Varna, Bulgaria).

### 2.2. Thin Film Characterization

The thickness measurement of the films was provided by an MProbe spectroscopic reflectometer MProbe UV-VisSR system developed by Semiconsoft Inc. (Southborough, MA, USA). Spectroscopic reflectometry is a powerful technique that can be used for a wide range of applications to measure both thickness and optical constants. The spectroscopic reflectance method has a clear advantage of higher precision of thickness measurements (with the exception of very thin films). The NiO_x_/Si samples were measured using the MProbe UV-VIS MSP system. Mprobe precision is ˂0.01 nm. The fit procedure for determing refractive index is less than 1% accurate.

A Shimadzu FTIR Spectrophotometer IRPrestige-21 (Shimadzu, Japan) was used to perform the Fourier Transform Infrared Spectroscopy measurements in the spectral range 350–4000 cm^−1^ with a resolution of 4 cm^−1^. The bare Si wafer was used as background.

The film composition and electronic structure were investigated by X-ray photoelectron spectroscopy (XPS). The measurements were carried out on an AXIS Supra electron- spectrometer (Kratos Analytical Ltd., Manchester, UK) using achromatic AlKα radiation with a photon energy of 1486.6 eV and a charge neutralization system. The binding energies (BE) were determined with an accuracy of ±0.1 eV. The chemical composition in the depth of the films was determined by monitoring the areas and binding energies of C 1s, O 1s, and Ni 2p photoelectron peaks. Using the commercial data-processing software of Kratos Analytical Ltd., the concentrations of the different chemical elements (in atomic %) were calculated by normalizing the areas of the photoelectron peaks to their relative sensitivity factors.

Optical transmittance and reflectance spectra were carried out by a UV–VIS–NIR Shimadzu 3600 double-beam spectrophotometer (Shimadzu, Japan) in the spectral region of 240–1200 nm with a resolution of 0.1 nm. The transmittance was measured against air and the reflectance was measured by using the specular reflectance attachment (5° incidence angle) and an Al-coated mirror as reference. The transmittance and reflectance spectra of bare glass substrate are given as reference.

The work function (WF) of the samples was measured with a Scanning Kelvin Probe (SKP5050). The equipment and methodology are from KP Technology, Wick, Caithness, Scotland, UK. The Kelvin probe measures the contact potential difference between a sample and tip. The WF of the sample is calculated by determining the WF of the tip calibrated against a known surface. The sheet resistance of NiO films on glass substrate structures was measured by the four point probe method using a VEECO instrument, model FPP-100, Plainview, New York, USA.

## 3. Results and Discussions

### 3.1. IR Study

IR spectroscopy is widely used for investigating sol-gel-derived films and the microstructural evolution with temperature and film processing parameters [25]. The FTIR spectra of NiO thin films are shown in Figure 2. The effect of annealing treatments on functional groups, vibrational modes, and the absorption band shapes and intensities was studied. It was seen that there are hydroxyl group vibrations that contribute to IR spectra and the corresponding absorption bands are changed with annealing. Generally, hydroxyl group stretching vibrations are mainly observed in the spectral range 3000–4000 cm^−1^ with the band at 1639 cm^−1^ attributed to the bending vibrations. The bands associated with the hydroxyl species are very strong and broad at lower annealing temperatures (200 and 300 °C), and they decrease in intensity with the increase of the temperatures to 400 and 500 °C.

The temperature treatment at 200 °C resulted in broad absorption bands centred at 3400 cm^−1^ for NiO_x_ films (Figure 2a). The broad absorption, covering from 3000 to 3680 cm^−1^, clearly indicates contributions of different O-H stretching vibrations. The band at around 3400 cm^−1^ is attributed to interlayer water molecules and H-bound OH [26], at 3520–3566 cm^−1^ is associated with hydroxyl groups connected with Ni(OH)_2_ [27], and the band at 3634 cm^−1^ is characteristic for Ni(OH)_2_ [28]. The bending OH vibrations can be seen around 1600 cm^−1^. The weak lines at 886 cm^−1^ and at 680 cm^−1^ are related to bending modes of OH lattice in Ni(OH)_2_ [27] and wagging δOH vibrations, respectively [29]. These absorption features are observed in the spectra of the 300 °C sample with diminished intensities. NiO_x_ films annealed at 400 and 500 °C manifest narrower and clear IR lines at 3615, 3750, and 3820 cm^−1^.

These lines are assigned to stretching vibrations of hydroxyl groups in free alignment [30] or due to Ni(OH)_2_ [27], interlayer water [31] and physisorbed H_2_O molecules, or terminal OH [32]. The peak at 880 cm^−1^ is changed, and at 675 cm^−1^, the line is shifted. IR lines at 2890, 2364, and 2334 cm^−1^ are due to C-H and to C-O (due to atmospheric CO_2_) vibrations [26], and respectively, the absorptions at 1420, 1350, 1065, and 1025 cm^−1^ are related to different C-H and C-O modes due to the deposition process [33]. The thermal treatments change the absorption bands due to hydroxyl groups, as the 3600 cm^−1^ band is shifted (300 °C) and vanished (for higher temperature annealed films).

The spectral range 350–650 cm^−1^ reveals many absorption lines attributed to Ni-O bonding vibrations. The line at 670 cm^−1^ (seen in all spectra, but very weak at higher temperatures) is related to hydroxyl groups in which H bonded to Ni-O [26]. This line suggests that adsorbed water is still present after annealing above 300 °C. The characteristic peaks of nanocrystalline NiO [34] are expressed at 445–540 cm^−1^, for all annealed films with the exception of NiO_x_, treated at 200 °C. The strong lines at 390–405 cm^−1^ (for films, treated at 300–500 °C) are assigned to transverse optical phonon frequency [35,36], observed in bulk NiO. The peaks at 470 and 520 cm^−1^ are due to Ni-O in cubic NiO [26] and to surface Ni-O [36], respectively. Interestingly, the lines at 380 and 460 cm^−1^ are associated with Ni(OH)_2_ [37], manifested in the spectra of NiO_x_ films treated at 200 and 300 °C.

FTIR analysis revealed that hydroxyl species and IR absorptions due to Ni(OH)_2_ are seen in all spectra, but they are strongly influenced by the annealing temperatures. The IR lines due to hydroxyl group vibrations are significantly decreased, and mostly disappear with thermal treatment. The presence of Ni-O bonding is proved for NiO_x_ films treated at temperatures 300–500 °C.

### 3.2. Chemical States of NiO_x_ Thin Films

X-ray photoelectronic spectroscopy (XPS) is applied for revealing the chemical composition, defects states, and bonding states of Ni and O in the sol-gel NiO thin films. NiO is a transition metal oxide, and it is important to know the oxidation states. NiO exhibits spin–orbit doublet and its multiple peaks result in a complicated line shape, peak asymmetries, complex multiplet splitting, shake-up, plasmon loss structure, and overlapping binding energies, and it is difficult to determine the proportion of different Ni valences [38,39]. Figure 3 presents the XPS spectra of Ni 2p, O 1s, and C 1s core levels and their fitted results using Gaussian curves after the subtraction of a Shirley baseline.

XPS study revealed that there was a carbon contimination, monitored in the C 1s region. The deconvolution of C 1s spectra of 200 and 400 °C annealed samples consists of two peaks. The peak at 284.9 eV is usually related to C-C and the second peak at higher binding energy (BE) at 288.3 eV assigned to O-C=O [40]. The presence of carbon impurities is expected, as FTIR study shows the presence of carbon oxide residuals and adsorption from air.

The individual deconvoluted core level O 1s XPS spectra of NiO_x_ films are presented in Figure 3b. The O 1s of NiO_x_ film, annealed at 200 °C, is resolved into two well-pronounced peaks at the binding energies of 532.3 eV and 535.1 eV. This result is intriguing as it clearly asserts that there is no formation of a NiO phase at this annealing temperature. The main peak (532.3 eV) is mostly assigned to carbon contamination, H-O-H (adsorbed water) [40,41,42], or to the presence of NiO(OH) [43,44] (with the corresponding Ni 2p3/2 peak at 856.4 eV) (see Figure 3c). The 535.1 eV BE peak can be ascribed to water vapor in the film structure [45]. The existence of hydroxides and water inclusions is confirmed by FTIR spectroscopy and the corresponding expressed absorption bands. It should be noted that the chemical analysis showed that the oxygen concentration was very high, at. 74.8% for this sample (see Table 1).

The annealing at a higher temperature (400 °C) drastically changed the film microstructure. The O 1s spectrum revealed two definite lines. The first main peak is at the binding energy of 529.5 eV, due to metal-oxygen (O-Ni^2+^) bonds [46], and it is referred to as almost pure stoichiometric NiO [38]. The other peak, at BE = 531.3 eV, has been attributed to the formation of Ni_2_O_3_ (O -Ni^3+^) by many authors [41,44,46] (the corresponding Ni 2p3/2 peak at 855.7 eV is detected as can be observed in Figure 3c, but this can be disputed as it is also associated with adsorption of different oxygen-containing species such as Ni(OH)_2_ and NIO(OH), and defects in NiO [47,48]). In our study, the possible contributions both of Ni_2_O_3_ and hydroxyl species cannot be ruled out. IR absorptions due to Ni(OH)_2_ and water inclusions are manifested.

The obtained Ni 2p spectra contain two spin–orbit coupling spectra, Ni 2p3/2 and Ni 2p1/2 peaks. Both of them can be separated into four (200 °C) and five (400 °C) components using the Gaussian function (Figure 3c). Most researchers focus their XPS spectra interpretation on Ni 2p3/2 spectra. The first conclusion drawn from the Ni 2p spectra is that there is no appearance of metallic Nio (with main line at BE = 852.6 eV) [49].

The Ni 2p spectrum of NiO_x_ film treated at 200 °C revealed two main peaks at 856.4 eV (Ni 2p3/2) and at 874.4 eV and two shake-up satellite peaks at 861.1 eV and 879.6 eV, respectively. As it has been discussed above, the peak at 856.4 eV is related to Ni^3+^ states in NiO(OH) [44], but other authors have assigned it to Ni_2_O_3_ (Ni^3+^) and to Ni(OH)_2_ (Ni^2+^) phases [46,50]. The probable existence of Ni hydroxides is supported by FTIR study and the positions of O 1s peaks.

The annealing at 400 °C leads to the appearance of a NiO phase as it has been seen from O 1s spectrum interpretation. The main peaks are located at 853.9 eV (Ni 2p3/2), 855.5 eV (Ni 2p3/2), and 872.8 eV (Ni 2p1/2). The first line, located at the binding energy of 853.9 eV, is attributed to Ni^2+^ states in NiO [40,42,50]. The Ni 2p3/2 at 853.9 eV and O 1s at 529.5 eV are reported to be due to Ni^2+^ states in Ni-O octahedrals in cubic NiO [43]. FTIR analysis revealed that the IR lines at 390–405, 470, and 520 cm^−1^ are due to Ni-O in cubic NiO for sol-gel NiO_x_ films, treated at 400–500 °C.

The peak with BE = 855.5 eV corresponds to Ni^3+^ states either in Ni_2_O_3_ [51] or NiOOH [42,50], and their assignments can be confirmed by O 1s spectra at BE = 531.3 eV. It has been reported that Ni^3+^ states result from quasi-localized holes around Ni^2+^ vacancies in the lattice sites, and this results in an increase of p-type conductivity [43,52].

XPS analysis manifested that NiOx films annealed at the lowest annealing temperature may consist of Ni hydroxides, and the formation of the NiO phase was proved after 400 °C treatment. The presence of Ni^3+^ states was also found. These conclusions are supported by FTIR data.

It is known that in its stoichiometric form, NiO is an insulator. NiO material, obtained by different methods, is a metal-deficient p-type semiconductor; therefore, Ni vacancies are present at the cation lattice site [9]. Due to Ni vacancies, some Ni^2+^ ions must be converted to Ni^3+^ ions in order to maintain the electrical neutrality in the structure [9]. The Ni^2+^ deficiency, Ni^3+^ presence, and oxygen-rich nature directly contribute to the p-type conductivity of the NiO_x_ film [53].

### 3.3. Thickness of the Films

The optical constants of NiO_x_ thin films depend on stoichiometry, deposition method used, technological conditions, and film structure. Thus, in many cases, it is imperative to measure n and k simultaneously with the thickness to obtain accurate results. The NiOx/Si samples were measured using an MProbe UV-VisSR system in the 200–1000 nm wavelength range. NiO_x_ dispersion was represented using the Exciton dispersion model, and T (transmittance), n (refractive index), and k (extinction) values were determined from the fit to the measured data. The results are presented in Table 2. A strong decrease of the film thickness was observed after thermal treatments. This can be due to significant densification for annealing temperatures above 200 °C. The hydroxyl groups and hydroxide and Nickel oxyhydroxide phases may exist in the film structure of 200 °C annealed samples, as has been deduced by IR and XPS measurements. Higher temperatures reduce these defects, and the NiO phase is formed.

The refractive index of bulk and stoichiometric NiO is 2.18 [54]. Generally, the crystalline material possesses a higher refractive index than the amorphous material. The values of measured refractive index for our sol-gel NiO_x_ films are in agreement with the reported data [55,56].

### 3.4. Optical Properties

Figure 4 shows the transmittance and reflectance spectra of NiO_x_ film in the spectral range 240–1200 nm as a function of the annealing temperatures. The substrate used was glass. The transmittance in the wavelengths from 450 to 800 nm was affected by the annealings, as it dropped from almost 90% (200 °C sample) to 74–77% of 500 °C treated NiO_x_ film. The reflectance reveals an increase from about 8% (200 °C) to about 20% (400 and 500 °C). It must be noted that the films cannot be directly compared due to significant differences in the film thickness of the 200 °C sample and the other NiO_x_ films.

The lower transparency and higher reflectance of NiO_x_ film annealed at temperatures above 200 °C can be due to structure modifications, crystallization, and rougher surfaces.

Figure 4 also manifests that the absorption edge of NiO_x_ films is dependent on the thermal treatments, and it is red shifted. The optical band gap of sol-gel NiO_x_ films is determined by Tuac’s plot, assuming direct transitions [1], namely the linear part of the (αhν)^2^ curve is extrapolated toward the energy axis at (αhν)^2^ = 0. The determined band gap values of NiO_x_ films are in Figure 5 as a function of the annealing temperatures.

The obtained results clearly reveal that there is shrinkage of the optical band gap of NiO_x_ films with increasing annealing temperatures. This can be due to the film crystallization, decreased defects of the films, and the increase of the crystallite sizes with the thermal treatments. Similar results have been reported in literature [1,10,40]. The decrease of the optical band gap and the shifting of the absorption edge to the higher wavelength with increasing annealing temperatures is reported for sol-gel NiO films [57]. This may be related to greater crystallites and decreased film porosity. The significantly smaller film thickness of the higher-temperature-treated NiO_x_ can be due to film densification.

Figure 6 shows spectral dispersion of the optical constants of sol-gel NiO_x_ films. The coefficient, *k* and refractive index, *n* of sol-gel NiO_x_ films have been determined from the spectrophotometric data using the following relations [55,58]:(1)$$\alpha =\frac{1}{d}In\left(\frac{1}{T}\right)$$
(2)$$k=\frac{\alpha \lambda }{4\pi },$$
(3)$$n=\left(\frac{1-R}{1+R}\right)+\sqrt{\frac{4R}{{\left(1-R\right)}^{2}}-k^{2}},$$
where *R* is the film reflectance (substrate reflectance is subtracted), λ is the wavelength of the incident beam, and α is the absorption coefficient. The low values of extinction reveal the good transparency of the studied films. The extinction coefficient of 200 °C treated film is below 0.03 (400–700 nm). The extinction coefficient is higher for NiO_x_ films treated at 300–500 °C, but their values are below 0.21 in the visible spectral range.

The refractive index is also dependent on the thermal treatments. It is known that the refractive index is higher for crystallized and denser samples than amorphous samples [58]. It can be seen that 200 °C film possess a refractive index of 1.24 at 630 nm and increases to 1.81 (300 °C), 2.02 (400 °C), and 1.86 (500 °C). There is a difference from the data given in Table 2, determined by spectral reflectometry. However, it must be pointed out that these values are measured of NiO_x_ films on Si substrate. The determined refractive index of the 200 °C annealed sample is very low, but its structure is complicated, as shown by FTIR and XPS studies. It includes Ni hydrooxides and oxyhydroxides and is presumably amorphous. Other authors [58] also found closer refractive index values for amorphous sputtered NiO films. Table 3 presents the values of refractive index and optical band gap for NiO thin films, deposted by different technological methods.

The values of n in the range of 1.45–2.16 for NiO_x_ films treated at 300, 400, and 500 °C and the optical band gap change (from 3.92 to 3.68 eV) coincide to the reported values as can be seen in Table 3 [54,59,60,61].

### 3.5. Electrical Characterization

We also determined the work function (WF) of NiO_x_ thin films, which is an indicator of the quality and composition of the layers. The WF of NiO_x_ thin films could be altered by different deposition techniques and the depositional parameters along with defects, such as oxygen vacancies or surface impurities, and composition [62]. It is reported that non-stoichiometric NiO_x_ films with high Ni vacancies and additional oxygen have a WF larger than 5.0 eV [63]. The work function of NiO can be altered by exposing the sample to air and annealing to induce defects and remove excess oxygen. Table 4 shows that the work function changes slightly with increasing annealing temperature (with the exception of the sample annealed at 200 °C). In this case, the surface nickel hydroxide and adsorbed carbonaceous species affect the work function [64].

The sheet resistivity of NiO_x_ films was measured by a four-point probe system, and the results are summarized along with annealing temperature in Table 5. The sheet resistivity of the NiO_x_ films was measured on the insulating glass substrates. This corresponds to a resistivity of about 4.8 × 10^−3^ Ω·cm for NiO_x_ film annealed at 400 °C. These values are significantly lower than those in [65,66,67].

Our results show that the samples after thermal treatment at 200 °C exhibit very low intrinsic conductivity. Increasing the temperature at and above 300 °C, NiO_x_ manifests p-type conductivity. As it has been mentioned above, some authors [10,68] claim that NiO_x_ is a metal-deficient p-type semiconductor and Ni vacancies are present at the cation lattice site, inducing the creation of Ni^3+^ ions. The metal vacancy and Ni^3+^ ions serve as hole acceptors, and this determines the p-type conductivity [69]. The presence of Ni^3+^ has been confirmed by XPS analysis for higher-temperature-treated NiO_x_ films.

## 4. Conclusions

It has been demonstrated in this work that p-type conductive NiO_x_ thin films of high quality can be deposited by the sol–gel technique using nickel acetate as a precursor instead of the usual alkoxides. The effect of the thermal treatments in the temperature range of 200–500 °C on the structural, optical, and electrical properties was studied. The main results can be summarized as follows:XPS analysis showed that NiO_x_ films annealed at the lowest annealing temperature may consist of Ni hydroxides and the formation of the NiO phase is proved after 400 °C treatment.XPS analysis revealed that the higher annealing results in the cubic NiO phase (Ni^2+^ states) with presence Ni^3+^ states, which contribute to p-type conductivity of the films.FTIR measurements also confirmed conclusions that the sol-gel NiO_x_ film, annealed at 200 °C, consisted of other Ni oxide phases not cubic NiO with presence of hydroxyl and carbon species.The transparency of NiO_x_ films is high, with a slight decrease with annealing with an extinction coefficient below 0.03 (400–700 nm) of 200 °C annealed film and below 0.21 for NiO_x_ films, treated at 300–500 °C in the visible spectral range. For NiO_x_ film annealed at 200 °C, the refractive index is found to be in the range of 1.18–1.33. For NiO_x_ film treated at 400 °C, it varies from 1.91–2.10. These values are determined for the spectral range 400–700 nm.The optical band gap energy of sol–gel NiO_x_ films decreases from 3.92 eV to 3.68 eV with increase in annealing temperature. These values coincide with the reported values for Ni oxide films.NiO_x_ thin films obtained by the sol-gel method are proven to possess p-type electrical conductivity. The 70-nm-thick layers have p-type conductivity and sheet resistance of 690 Ω/sq, which corresponds to a specific resistivity of about 4.8 × 10^−3^ Ω·cm. It was shown that at temperatures above 300 °C, NiO_x_ thin films with good optical and electrical properties are obtained. Based on the research of the optical and electrical properties of NiO_x_, it was found that the quality of the NiO_x_ films is comparable to that of films prepared by more complicated and expensive techniques such as rf magnetron sputtering, electron beam evaporation, MBE, etc.The advantages of the sol-gel process are low cost, easy fabrication, long-term stability, and that they are suitable for use in large areas. The applied technological approach is compatible with that of deposition of zinc oxide, which implies successful integration of sol-gel processes for the realization of visible-light transparent solar cells.

## Figures and Tables

**Figure 1 materials-15-01742-f001:**
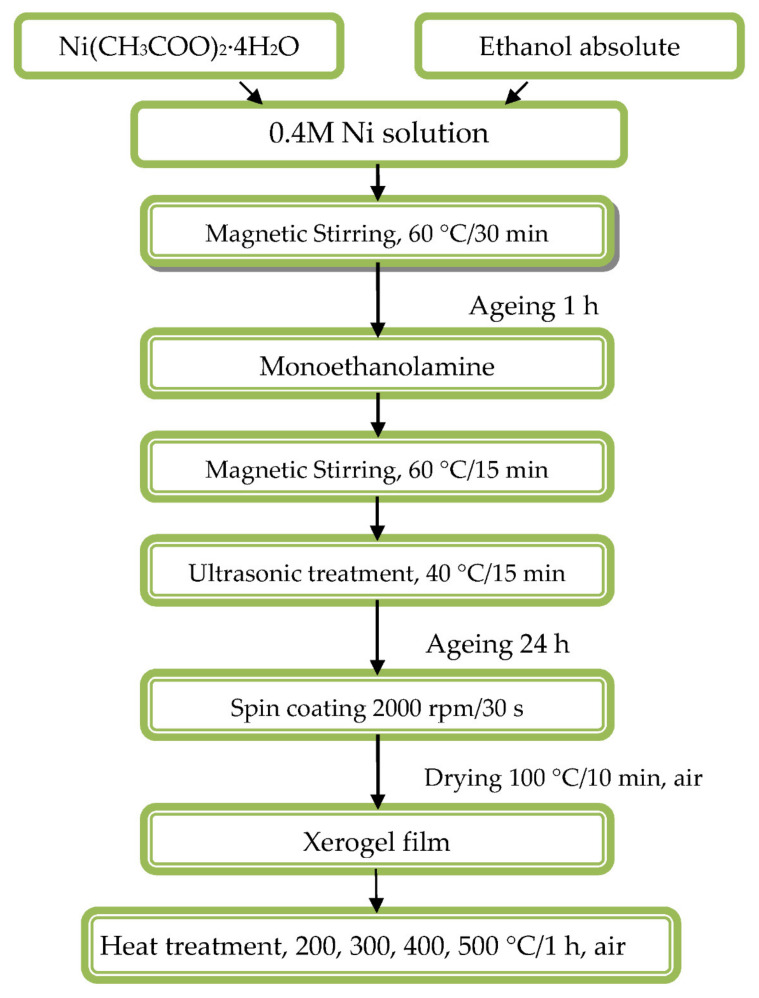
Scheme of the sol solution preparation and technological sequence of NiO_x_ film deposition.

**Figure 2 materials-15-01742-f002:**
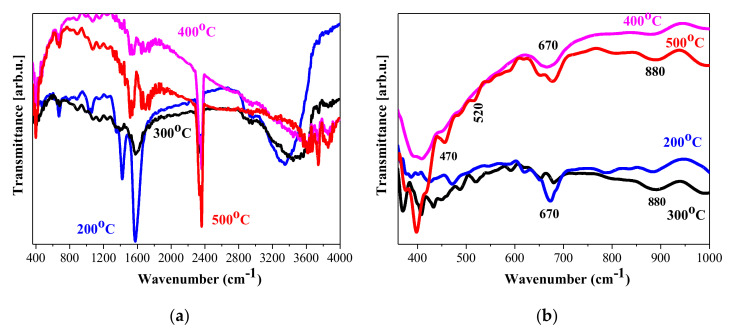
(**a**) FTIR spectra of sol-gel NiO_x_ films treated at 200 °C, 300 °C, 400 °C, and 500 °C; (**b**) presents FTIR spectra in the shorter spectral range 350 - 1000 cm^-^^1^.

**Figure 3 materials-15-01742-f003:**
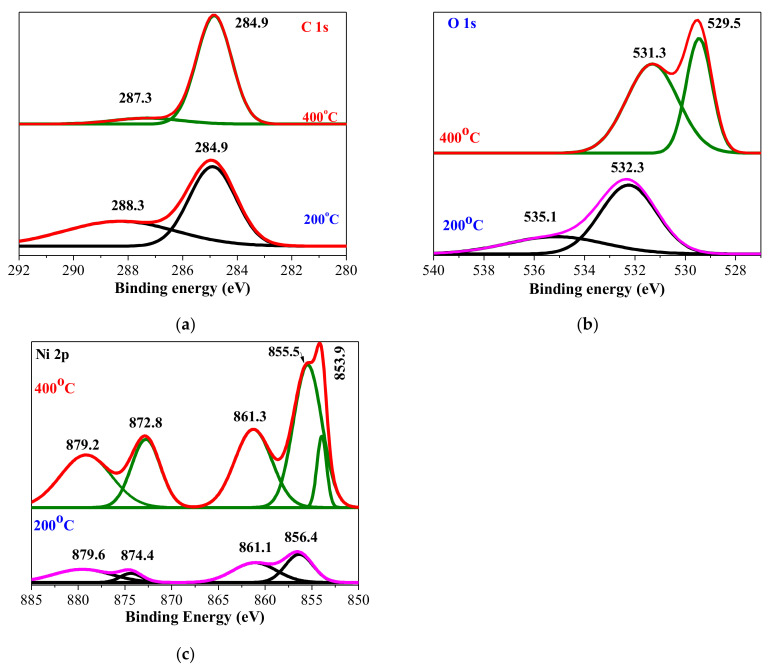
XPS spectra of (**a**) C 1s, (**b**) O 1s, and (**c**) Ni 2p core levels of NiO_x_ films, 1 layer, treated at 200 °C and 400 °C.

**Figure 4 materials-15-01742-f004:**
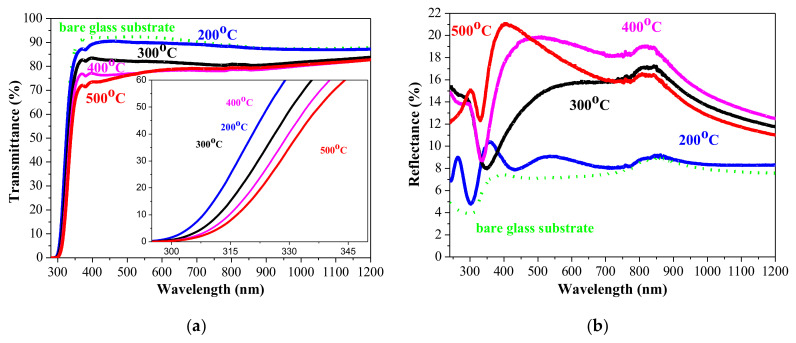
(**a**) Transmittance spectra of NiO_x_ films, annealed at the temperatures of 200, 300, 400, and 500 °C. The inset figure presents the absorption edges of NiO_x_ films. (**b**) Corresponding reflectance spectra.

**Figure 5 materials-15-01742-f005:**
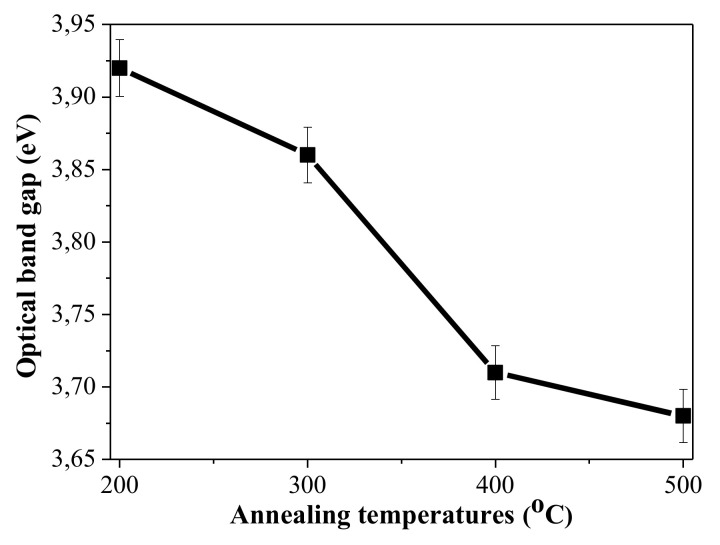
Estimated optical band gap values of sol-gel NiO_x_ films depending on the annealing temperatures from 200 to 500 °C.

**Figure 6 materials-15-01742-f006:**
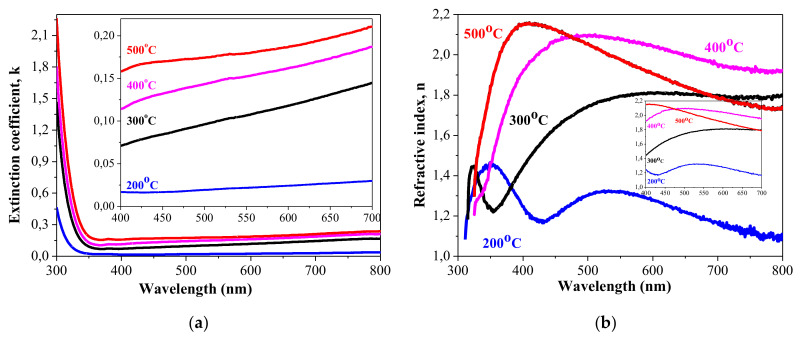
Extinction coefficient (**a**) and refractive index (**b**) of sol-gel NiO_x_ films.

**Table 1 materials-15-01742-t001:** The surface chemical composition of the sol-gel NiO_x_ films.

Component	Atomic %	Mass Conc. %	Ni/O Atomic %
NiO_x_ films, annealed at 200 °C			
O 1s	74.8	48.3	0.27
Ni 2p	20.0	47.4	
NiO_x_ films, annealed at 400 °C			
O 1s	56.4	26.1	0.76
Ni 2p	42.9	72.9	

**Table 2 materials-15-01742-t002:** Film thickness and refractive index determined at the wavelength of 630 nm for sol-gel NiO_x_ films. The NiO_x_/Si samples were measured using an Mprobe UV-VIS MSP system with precision <0.01 nm. The fit procedure for determing refractive index is less than 1% accurate.

Annealing Temperature [°C]	Film Thickness [nm]	Refractive Index NiOx/Si, Reflectometry	Refractive Index NiOx/Glass, from R, T Data
200	213	1.55	1.24
300	81	1.86	1.81
400	73	1.90	2.02
500	62	1.90	1.86

**Table 3 materials-15-01742-t003:** The optical band gap and refractive index values of NiO films, deposited by various methods.

Optical Band Gap [eV]	Refractive Index	Spectral Range	Deposition Method	Reference
3.572	2.15	Average, 280–900 nm	Spray pyrolisis	[1]
3.63	-	-	Thermal evaporation	[10]
3.40		(aqueous solution)	Chemical spray	[40]
4.00		(alcoholic solution)		
3.25–4.00	1.871	550 nm	Spray pyrolisis	[54] review
3.78–3.85	˃2	450–800 nm	Sol-gel	[55]
3.73	1.82–1.42	300–800 nm	Sol-gel	[56]
3.47–3.86	Decrease with annealing		Sol-gel	[57]
3.12–3.93	2.0–1.4	300–1000 nm	Magnetron sputtering	[58]
3.954	2.2–1.9	400–1000 nm	E-beam technique	[59]
3.14, 3.83	2.8–1.8	300–1000 nm	Spray pyrolisis	[60]
3.50–3.81	2.10–1.45	300–1000 nm	Wet method	[61]

**Table 4 materials-15-01742-t004:** Work Functions of sol-gel NiO_x_ films, treated at different annealing temperatures.

Annealing Temperature [°C]	Workfunction [eV]
200	4.44
300	5.29
400	5.25
500	5.24

**Table 5 materials-15-01742-t005:** Sheet resistance values of sol-gel NiO_x_ films treated at temperatures from 200 to 500 °C.

Annealing Temperature [°C]	Sheet Resistance [Ω/sq]
200	-
300	689
400	690
500	689

## Data Availability

The data are not publicly available.

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
