# Peer review of "Nickel Oxide Films Deposited by Sol-Gel Method: Effect of Annealing Temperature on Structural, Optical, and Electrical Properties"

_materials, 2022, doi:10.3390/ma15051742_

Round 1
Reviewer 1 Report
NiOx thin films are of great interest for applications including electrochromic windows, UV photodetector, and solar cells. These applications rely heavily on the p-type conductivity of the NiOx thin films, while stoichiometric NiO is an insulator with high resistivity. Therefore, enhancing the electrical conductivity of NiO films is a critical and interesting topic to improve the applicability of NiO films. The manuscript describes the deposition and properties of NiOx thin films by Sol-Gel process. The results show that the films have high optical transmittance and electrical conductivity. Detailed study on the films’ optical, structural, and electrical properties are carried out and presented. The conclusions are supported by data from different metrologies. However, the manuscript requires major revisions to address the questions that follow.
Major issues,
- The structure of the introduction needs to be organized in a better way. Paragraph 1-3 describe the attractive properties of NiOx, especially those of the non-stoichiometric oxide films, for various applications. Paragraph 4 starts to describe the advantages and disadvantages of the existing deposition methods of NiO films. Yet starting from the 5th paragraph, more application areas for NiOx films are introduced. The paragraphs should be rearranged in a coherent way so that the readers could follow the line of reasoning smoothly.
- The motivation of this work should be highlighted to establish the significance of this work. It is stated that the ‘aim of this work is to achieve a technological process for deposition’ of NiOx thin films. More description on this topic, such as those at the end of the manuscript (‘process is compatible with that of zinc oxide, which implies successful integration of sol-gel processes for the realization of visible-light transparent solar cells’), should be presented in the introduction part.
- The tense used in the description of experimental details should be consistent and in past tense.
- Most of the data presented in this work such as film thickness and optical band gap are missing experimental error bars.
- Description of optical properties in section 3.4 should be moved forward after section 3.1 so that the refractive indexes obtained from the two methods could be compared and discussed.
- The discussion in line 305 does not provide a clear explanation why the refractive indexes of the sample annealed at 200°C obtained from the two methods are different. The discussion seems to indicate that it’s due to the different substrate used. The film property, however, should remain similar. Or the method requires significant refractive index difference between the film and the substrate?
- Line 293, how absorption coefficient is calculated is missing.
- The first paragraph in section 3.2 gives a broad view of how the IR spectra change with annealing. The band range should be given for the hydroxyl species.
- Line 168, the conclusion that ‘the peak at 880 cm-1 is vanished’ (after annealing?) is hard to draw. Peak indexing in the Figure 1 (b) should be included to guide the reader to track the change in peak intensity with annealing.
- Line 175, the peak is not centered at 650 cm-1 and it’s hard to conclude that the peak is weaker at higher temp.
- The statement should be clearer that the resistivity of 4.8×10-3 Ω·cm reported in line 329 is the case for the sample annealed at 400°
- The 4PP results show that the sample annealed at 200°C has high resistance that’s not measurable. Did the authors observe any charging effect in XPS measurement for this sample? Was neutralizer used during the XPS measurement?
- There are a significant number of typos in the manuscript. The authors should address them and review the work more carefully.
Minor issues,
Line 25, ‘semiconductor’ to semiconductors
Line 113, degree symbol has an underline
Line 130, ‘it is observed that’ there’s ‘a strong decrease’
Line 132, ‘hydroxide’ to hydroxide
Line 133, ‘maybe’ to may
Line 136, remove the comma
Line 137, in table header, ‘reflectrometry’ to reflectometry
Line 158, ‘as’ to peak at
Line 196, missing space between s and core
Line 225, ‘Figure C)’ to Figure 2(C)
Line 230, remove spectra
Line 234, ‘Ni0’ to Ni0
Line 239, ‘supposed’ to supported?
Line 245, remove ‘e’ before ‘due to’
Line 269, ‘significant’ difference in ‘film thickness’?
Line 276, ‘manifest’ to ‘manifests’
Line 280, ‘as a function of annealing’ temperature
Line 285, remove ‘the’ between ‘increasing’ and ‘annealing temperature’
Line 291, the phrase is confusing and need re-work
Line 292, Figure 5 is missing a caption
Line 297, numbers need to be given for ‘low values of extinction’ coefficient
Line 299, the last sentence is redundant as the first sentence in the next paragraph is making the same statement
Line 324, remove the comma after ‘sample’
Line 351, ‘slightly’ to slight
Line 355, ‘resistance’ to resistivity
Author Response
Please, see the attachment

Reviewer 2 Report
The manuscript investigated the Deposition and Properties of NiOx Thin Films by Sol-Gel Process which conducted a number of characterizations for the developed thin films that showed promising results. However, the article needs significant improvement to get published in Materials
Few comments are provided below:
- The title is general and there are similar titles in the literature. Try to make it more specific and directed to the work conducted.
- The introduction has many isolated paragraphs that need to be combined and connected.
- The Materials and Methods section needs to be divided into subsections for example for thin film production as a subsection and Characterization as another subsection
- Initially describe the obtained film thickness relative to the reported in the literature and also the required thickness for a specific application.
- The reduction of the film thickness needs to be scientifically justified not just mentioning it is maybe due to so and so.
- Please add the accuracy of measurement for the thin films
- “The obtained Ni sol solution is 76 found to be homogeneous, transparent without precipitations and with good film 77 forming properties.” Have you made any analysis for this Ni sol solution as this represents the starting material for the films formed?
- It is important to mention both figures and tables before they appear in the text as much as possible.
- The presentation of Figure 2 needs to be adjusted and improved.
- Figure 5 on page 8 has no number and figure caption
- The conclusions section needs to be rewritten and restructured. The main finding of the work should be given clearly in bullet points.
- Although the number of references used is 67 only 22 of them are covered in the introduction. While the introduction needs improvement.
Author Response
Please, see the attachment.

Reviewer 3 Report
The manuscript reports the deposition NiOx sol-gel films on glass and Si substrates. The manuscript is within the scope of Journal materials. However, before publication some major concerns, namely:
- The abstract must be improved and must be turned more appealing as well as reflect the most interesting results and conclusions of the work herein reported.
- The introduction must be improved. For instance, the authors state that "Due to these unique properties, NiOx thin films become multipurpose materials in various applications such as electrochromic (EC) applications [6]." and then only one application example is provide. The following sentences discuss some examples, however, do not provide data that allows to have the big picture of the actual state of the art. Please discuss the most promising applications of these films already reported.
- Section 2 - Please introduce a schematic of the synthesis steps.
- Table 1 - Please introduce the errors linked to the thickness measurements.
- Also a space between the number and the units must be used. For instance, this was not done when temperature values were used.
- The authors must introduce a table with the results already reported to compare to the results reported herein. Since this will allow an easy analysis and comparison. As presented, it is difficult to the readers compare the results reported with the ones already available in the literature.
- In the conclusions some of the units are not with the correct symbols (i.e 690 Ω/â–¡).
Author Response
Please, see the attachment

Round 2
Reviewer 1 Report
The revised manuscript is acceptable for publication in its present form.
Reviewer 2 Report
Although the changes made in the manuscript are not highlighted by any means, the majority of the provided comments were considered and the article improved, however, a few comments still needs more attention by the authors before acceptance:
Line 201-202 “ The individual deconvoluted core level O1s XPS spectra of NiOx films are presented in Figure 2 b. The O 1s of NiOx film, annealed at 200 oC is resolved into two” Correct the Figure no it seems this is Figure 3b. Revise the use of figure numbers throughout.
The conclusion section still needs to be rewritten in bullet point clear outcomes of the work supported with results.
Reviewer 3 Report
All the issues raised were correctly addressed.
